# GCase Enhancers: A Potential Therapeutic Option for Gaucher Disease and Other Neurological Disorders

**DOI:** 10.3390/ph15070823

**Published:** 2022-07-02

**Authors:** Macarena Martínez-Bailén, Francesca Clemente, Camilla Matassini, Francesca Cardona

**Affiliations:** 1Dipartimento di Chimica “Ugo Schiff” (DICUS), Università degli Studi di Firenze, Via della Lastruccia 3-13, 50019 Sesto Fiorentino, Italy; francesca.clemente@unifi.it (F.C.); camilla.matassini@unifi.it (C.M.); 2Glycosystems Laboratory, Instituto de Investigaciones Químicas (IIQ), CSIC—Universidad de Sevilla, Av. Américo Vespucio 49, 41092 Sevilla, Spain

**Keywords:** Gaucher disease, lysosomal storage disorders, pharmacological chaperones, allosteric chaperones, iminosugars, Parkinson disease

## Abstract

Pharmaceutical chaperones (PCs) are small compounds able to bind and stabilize misfolded proteins, allowing them to recover their native folding and thus their biological activity. In particular, lysosomal storage disorders (LSDs), a class of metabolic disorders due to genetic mutations that result in misfolded lysosomal enzymes, can strongly benefit from the use of PCs able to facilitate their translocation to the lysosomes. This results in a recovery of their catalytic activity. No PC for the GCase enzyme (lysosomal acid-β-glucosidase, or glucocerebrosidase) has reached the market yet, despite the importance of this enzyme not only for Gaucher disease, the most common LSD, but also for neurological disorders, such as Parkinson’s disease. This review aims to describe the efforts made by the scientific community in the last 7 years (since 2015) in order to identify new PCs for the GCase enzyme, which have been mainly identified among glycomimetic-based compounds.

## 1. Introduction

Lysosomal storage disorders (LSDs) are a group of more than 70 genetic metabolic diseases characterized by lysosomal dysfunction due to gene mutations that encode for lysosomal enzymes. The gene mutations affect the function of the encoded protein, which is not correctly folded and thus undergoes endoplasmic reticulum-associated protein degradation (ERAD). As a consequence, the protein-misfolding phenomenon causes a lack of catalytic activity, leading to accumulation of unmetabolized substrates in the lysosomes [1,2].

Pharmacological chaperones (PCs) are small compounds able to bind and stabilize misfolded lysosomal enzymes, facilitating their translocation to the lysosome, which results in the recovery of catalytic activity. The great majority of PCs identified to date bind to the catalytic site acting as inhibitors, but once the enzyme arrives at the lysosome, the PC dissociation permits substrate turnover. These inhibitors act as a template, stabilizing the native folding state in the endoplasmic reticulum (ER) by occupying the active site of the mutant enzyme, with a consequent final counterintuitive effect of enhancing lysosomal enzyme catalytic activity [3,4] (Figure 1). However, more recently, non-competitive inhibitors acting as PCs have also been discovered.

Gaucher disease (GD) is the most common LSD of glycosphingolipids and originates from mutations in the *GBA* gene (chromosome 1q21−22), which encodes for the lysosomal enzyme acid-β-glucosidase (glucocerebrosidase or GCase, EC 3.2.1.45, MIM*606463). GCase is the enzyme responsible for the hydrolysis of glucosylceramide (GlcCer) to glucose and ceramide in the lysosomes. Due to its heterogeneity, GD is classified into three types: type 1, primarily not associated with neurological symptoms, and neuronopathic types 2 and 3 [5].

More recently, a connection between GCase activity and onset of Parkinson’s disease (PD), the second most common neurological disorder, was highlighted. GCase mutations were shown to be a major risk factor for the onset of PD, since people carrying heterozygous mutations in the *GBA* gene do not develop GD, but have an increased risk of developing PD [6,7,8]. More interestingly, the loss of GCase function contributes to the pathogenesis of PD, even independently of the genetic mutation [9], suggesting that modulation of GCase activity may be the key therapeutic target for both pathologies.

Over the past 20 years, pharmacological chaperone therapy (PCT) has been investigated as a potential treatment for LSDs, including GD, and has been the subject of numerous reviews [3,10,11,12,13,14,15,16,17,18,19,20]. However, few of these have been limited to GD, i.e., also include other LSDs, and do not cover recent years. Moreover, since the GD–PD connection emerged only recently, the potential use of PCT for treating PD has not been extensively described before.

Iminosugars are among the most studied classes of compounds as potential PCs for GD, and include structural classes belonging to polyhydroxylated pyrrolidines, piperidines, pyrrolizidines, indolizidines, and nortropanes [21]. In particular, isofagomine (IFG, **1**, Figure 2) is the compound that reached the most advanced clinical trial, but this stopped at phase II in 2018. IFG is a competitive inhibitor of human lysosomal GCase (*K*_i_ = 0.016 ± 0.009 µM; IC_50_ = 0.06 µM) [22], and was found to increase mutant GCase activity 1.6-fold at 10 µM after incubation in fibroblasts bearing homozygous N370S missense mutations (Figure 2) [23]. However, IFG was not effective in reducing the accumulation of lipid substrates when administered to adult patients with type 1 GD, probably due to its high hydrophilicity, which hampers efficient transport to the cells. To overcome this issue, several alkylated piperidine iminosugars have been proposed.

6-Nonyl IFG (**4**) [24,25], α-1-*C*-nonyl-DIX (**5**) [26], and *N*-nonyl-deoxynojirimycin (NN-DNJ, **6**) [27,28] were among the most effective chaperones identified for GD until 2015, being able to strongly enhance the mutated GCase activity in N370S GD fibroblasts, as shown in Figure 2. It is worth noting that the position of the alkyl chain is crucial for the pharmacological chaperoning activity dependent on the peculiar hydroxypiperidine skeleton. Indeed, while for deoxynojirimycin (DNJ, **3**) analogues, *N*-alkylation renders good PCs for GCase (e.g., compound **6**, Figure 2), in the case of IFG (**1**) and 1,5-dideoxy-1,5-iminoxylitol (DIX, **2**) analogues, the alkylation of the carbon adjacent to nitrogen furnishes better PCs for GD (e.g., compounds **4** and **5**, Figure 2). Despite these encouraging results, no PC for the GCase enzyme has reached the drug market yet.

This review focuses on the efforts made by the scientific community in the last 7 years (since 2015) in order to identify new PCs for GCase. Both inhibitory compounds, historically the first pharmacological chaperones identified, and allosteric enhancers will be discussed in this account. While the former, covered in the first section, are lysosomal enzyme inhibitors with a counterintuitive effect at sub-inhibitory concentrations corresponding to chaperone behavior, the latter are commonly discovered through a high-throughput screening approach or drug repurposing, and will be the subject of the second section. The final effect of both types of PC is an increase in enzymatic activity. We have chosen to organize the inhibitory chaperone section according to compound structure: iminosugars (six-membered, five-membered, bicyclic, and multivalent iminosugars), sugar analogues, and miscellaneous examples. Although a huge number of strong GCase inhibitors have been reported, the discussion will be limited to those inhibitors that have been tested in vitro on mutated human cells.

The second section, which is more innovative in its conception, will cover GCase enhancements and mechanistic studies relative to the very few and cutting-edge examples of allosteric chaperones.

The last section of this review is devoted to relevant examples of GCase chaperones assayed in PD animal models.

## 2. Inhibitory Chaperones

### 2.1. Iminosugars

#### 2.1.1. Six-Membered Iminosugars

NN-DNJ (**6**, Figure 2) was reported as the first iminosugar with potential for the treatment of GD [27]. Since then, further investigation has been performed in the search for piperidine-derived compounds with potential applications as chaperones for GD. Parmeggiani et al. [29] developed an iminosugar-based library composed of polyhydroxylated piperidines, pyrrolidines, and pyrrolizidines (see Section 2.1.2 and Section 2.1.3 for pyrrolidine and pyrrolizidine analogues, respectively). Regarding the piperidine derivatives, they were prepared from a carbohydrate-derived aldehyde and incorporated different moieties, such as a lipophilic chain or a benzyl group, either on the endocyclic or the exocyclic nitrogen atom. More polar groups (aminomethyl, carboxylic acid, or polar-ending lipophilic chain) were also investigated. When evaluated towards GCase, the best inhibitory activities were obtained for derivatives **7** and **8** bearing an eight-carbon alkyl chain (IC_50_ = 30 and 40 μM, respectively). Both compounds showed a measurable chaperoning activity towards human fibroblasts derived from Gaucher patients bearing the N370S/RecNcil mutations (1.25- and 1.5-fold enzyme-activity enhancement at 100 μM, respectively) (Figure 3).

A previous study pointed out that sp^2^-iminosugars behave as strong competitive inhibitors of GCase and have potential as PCs for GD [30]. Mena-Barragán et al. [31] reported the synthesis of several DNJ-based sp^2^-iminosugars incorporating an orthoester fragment to switch the nature of the aglycone moiety from hydrophobic to hydrophilic in the pH 7 to pH 5 window, which had a dramatic effect on enzyme-binding affinity, thus maximizing the chaperone over the inhibitory behavior. The authors chose the orthoester group due to its chemical acid lability and stability in neutral conditions, properties that allow for optimal performance in a useful pH window. Compounds **9**–**11** (Figure 4) behaved as selective GCase inhibitors and were 1.3- to 200-fold better ligands of this enzyme than ambroxol (IC_50_ = 41.5 μM, see Section 2.3), a non-glycomimetic PC under clinical trial for Gaucher disease [32], at neutral pH in the endoplasmic reticulum (ER). At an acidic pH (lysosome), the product **12** (Figure 4) obtained from the hydrolysis of **9**–**11**, was 3-fold weaker ligand than ambroxol.

When tested in cell patients with the N188S/G193W mutation associated with the type 3 phenotype of GD, compound **9**, bearing a butoxy chain, showed better enhancing activity (6-fold activity increase at 50 μM) than the octyl and dodecyl analogues **10** and **11**. Compound **9** additionally proved to be efficient in enhancing GCase activity in V230G/R296X or F213I/F213I type 2 GD fibroblasts. However, in type 1 GD fibroblasts (N370S/N370S), the enhancement was more modest (1.5-fold enhancement). In the case of L444P/L444P fibroblasts, no activity rescue was observed.

Apart from DNJ (**3**, Figure 2), other six-membered glycomimetics, such as DIX (**2**, Figure 2), were studied by Sevsěk et al. [33] for the preparation of guanidine (**13**) and urea (**14**) DIX-derived compounds (Figure 5) bearing different lipophilic chains at the exocyclic nitrogen atom. The prepared compounds were evaluated against a panel of commercial glycosidases and human recombinant enzymes: GCase and β-galactocerebrosidase (GALC). Regarding the results for GCase, only the guanidine derivatives were potent and selective inhibitors in the low nanomolar range (IC_50_ = 17–245 nM), probably due to the positive charge of the guanidinium group, as suggested by the authors. The guanidine derivatives were better inhibitors than the simple *N*-nonyl DIX (IC_50_ = 752 nM), and this result agrees with the docking studies performed by the authors, which highlighted additional cation–π interactions and extra hydrogen bonding in the guanidine function. The length of the lipophilic chain in the molecule also affected the inhibition: compounds with longer chains were better inhibitors. The authors carried out experiments using fibroblasts homozygous for the N370S mutation, and compound **13a** (Figure 5) displayed measurable chaperone activity (1.2-fold at 1 μM).

Génisson and coworkers reported how ring expansion of pyrrolidine derivatives (**15** and **17**, Figure 6) to the piperidine ring system (**16** and **18**, Figure 6) led to a selective targeting of GlcCer formation or hydrolysis. The authors identified in *C*-octyl piperidine **16** and 2-*O*-heptyl piperidine **18** good GCase inhibitors (IC_50_ = 11 nM and 4 nM, respectively) and PCs with low cytotoxicity. A remarkable rescue of 3-fold GCase activity was observed at 100 nM for both compounds **16** and **18** after incubation (3 days) with homozygous N370S GD fibroblasts. On the contrary, the five-membered octyl derivatives **15** and **17** showed inhibitory activity towards the glucosylceramide synthase enzyme, involved in the synthesis of GlcCer, but behaved as weaker GCase inhibitors than their corresponding six-membered counterparts (Figure 6). Molecular docking studies showed that piperidines **16** and **18** bound the active site similarly to isofagomine (IFG (**1**), Figure 2) [34].

The role of alkyl chain length in six-membered glycomimetics on PC activity was also investigated by Cardona and coworkers, who reported the synthesis of trihydroxypiperidines **19** and **20** (Figure 7) alkylated at C-2 with both configurations and chains of different lengths [35,36]. The new compounds, assayed towards GCase and other lysosomal enzymes, proved to be good and selective GCase inhibitors with IC_50_ in the micromolar range. Selectivity towards GCase over other lysosomal enzymes is important to avoid undesired side effects. An interesting trend in the inhibitory activity was highlighted, which was dependent on the configuration at C-2 and chain length. In particular, piperidine **20d** (Figure 7) was the best inhibitor among the whole series, with an IC_50_ = 1.5 μM.

All compounds were able to rescue the activity of mutated GCase from 1.2- to 1.9-fold in fibroblasts derived from Gaucher patients bearing the N370S/RecNcil mutations. The best GCase rescue was obtained with the octyl compound with the *R* configuration at C-2 **20a** (Figure 7), which was able to enhance GCase activity up to 1.9-fold at 50 µM in fibroblasts bearing the N370S mutation. Remarkably, in the presence of compound **20a**, an 80% activity rescue (1.8-fold GCase enhancement) was measured in fibroblasts bearing the homozygous L444P mutation, which is often resistant to most PCs. With a longer chain, compound **20d** was the best inhibitor of the series, but not the best pharmacological chaperone. This effect can be explained in light of the fact that during in vitro assays on cell lines, other factors play a crucial role, such as cell permeability and cell viability. The improved potency provided by a longer alkyl moiety was also concurrent with low cell viability. These data demonstrated, analogously to what happens with IFG and DIX derivatives, that the presence of a lipophilic chain of proper length plays a pivotal role on the chaperoning properties and that the alkylation of the carbon adjacent to nitrogen furnishes better PCs for GD also in the case of these trihydroxypiperidine iminosugars.

#### 2.1.2. Five-Membered Iminosugars

Five-membered iminosugars (polyhydroxylated pyrrolidines) are a group of glycomimetics that—in comparison with their six-membered analogues—have been less studied in the field of glycosidase inhibition. This family of compounds sometimes shows problems of selectivity due to their higher conformational flexibility with respect to piperidines, but the attachment of different functional groups to the pyrrolidine core can lead to more potent and selective glycosidase inhibitors [37].

Using a d-arabinose-derived nitrone as key intermediate, Parmeggiani et al. [29] developed a library of polyhydroxylated pyrrolidines, analogues of the naturally occurring iminosugar 1,4-dideoxy-1,4-imino-d-arabinitol (DAB, **21**, Figure 8), in their *N*- or *C*-alkylated version (chain length of 4, 8, or 12 carbon atoms). The best results in terms of GCase inhibition were obtained with pyrrolidines bearing a longer chain (8 or 12 atom carbons, compounds **22**–**27**, Figure 8), thus probably suggesting that these compounds better mimic the natural substrate of GCase. Unfortunately, despite the good inhibitory activity displayed for compounds **22**–**27** (IC_50_ = 0.72–150 μM, Figure 8), they did not show any chaperone activity when tested on cell lines.

Conversely, Kato et al. [38] prepared DAB (**21**, Figure 8) and LAB (1,4-dideoxy-1,4-imino-L-arabinitol, ***ent*-21**, Figure 9) *C*-alkylated pyrrolidines bearing a butyl or octyl chain (compounds **23**, **28**, ***ent*-23** and ***ent*-28**, Figure 9). The inhibition results towards GCase showed that the best candidates were the DAB-configured analogues, and among them, the insertion of an octyl group (**23**) gave the best inhibitory activity. So, in a next step, the length of the alkyl chain was changed, ranging from 4 to 13 carbon atoms (derivatives **27** and **29**–**35**, Figure 9). The results showed that the longer the alkyl chain is, the stronger is the inhibition, with optimal results obtained for the tridecyl analogue **35** with an IC_50_ value of 0.77 μM. The derivatives **29**, **32**, and **35** were evaluated as chaperones in fibroblasts with the homozygous N370S mutation and compared with isofagomine (IFG (**1**), Figure 2), used as a positive control. Compound **35** was the best chaperone candidate, with a GCase rescue similar to that observed with IFG (around 1.5-fold at 5 μM) but at a 10-fold lower concentration (0.5 μM).

The presence of the hydroxymethyl moiety in the pyrrolidine skeleton makes an important contribution to glycosidase inhibition [39], but its removal can lead to more hydrophobic compounds that can better cross the cell membranes. Indeed, Castellan et al. [40] reported the synthesis of a couple of enantiomeric pyrrolidines, **36** and ***ent*-36** (Figure 10), and their evaluation as GCase inhibitors and activity enhancers in fibroblasts bearing the homozygous N370S mutation. These compounds can be also considered DAB (**21**, Figure 8) and LAB (***ent*-21**, Figure 9) analogues, respectively, but lacking the hydroxymethyl moiety. The removal of this group was advantageous for GCase inhibition, as both compounds were more potent inhibitors than the corresponding hydroxymethylated counterparts reported by Kato et al. (compounds **36** vs. **23** and ***ent*-36** vs. ***ent*-23**) [38].

Compound **36**, with a non-competitive inhibition mode, was two orders of magnitude more potent than competitive ***ent*-36**. The inhibitory mode of both compounds was further confirmed by the application of Theorell’s graphical method [41] and through docking simulations, which showed a tight alignment of ***ent*-36** with NN-DNJ (**6**, Figure 2) and IFG (**1**, Figure 2) and the retention of the main hydrogen-bond interactions within the GCase active site.

The chaperone activity of both compounds was measured in fibroblasts with the homozygous N370S mutation. Compound **36** gave the best results, showing an activity enhancement comparable to that observed for NN-DNJ (**6**) and IFG (**1**), used as reference compounds, i.e., 1.5-fold or 2-fold at 3 μM independently of the chosen cell lines. Indeed, it is worth noting that intracellular enzymatic enhancement may highly depend on the chosen cell lines, given the same N370S mutation. Nevertheless, the enhancement of ***ent*-36** (1.6-fold at 3 μM) is also significant, despite being a less potent competitive inhibitor (IC_50_ = 59.6 μM), demonstrating that not always the more potent inhibitors are also the best chaperone candidates.

Not only the insertion of alkyl chains on the pyrrolidine core can improve the biological properties towards the GCase enzyme. The introduction of (hetero)aromatic moieties exploiting copper(I)-catalyzed alkyne-azide cycloaddition (CuAAC) can also afford more potent and selective GCase inhibitors [42], probably due to the establishment of additional interactions with the enzyme, allowing at the same time improved cell permeability and cell trafficking. In this context, Martínez-Bailén et al. [43] reported the synthesis of a set of epimeric dihydroxypyrrolidines containing an aromatic moiety attached to the triazole generated in the CuAAC reaction. This new family of inhibitors exhibited a tunable biological activity towards GCase in the function of the substitution pattern of the aromatic ring, the 3,5-disubstituted derivatives being the most potent inhibitors (compounds **37**–**56**, Figure 11). The best inhibitors of the library (compounds **44**, **45**, and **53**) were able to increase the in vitro activity of GCase in Gaucher patient fibroblasts with the homozygous N370S mutation. Specifically, compound **45** increased the GCase activity 2-fold at 10 μM concentration, an enhancement that is comparable to that observed for isofagomine used as positive control at the same concentration. The crystal structure of complex **53**:GCase and docking experiments revealed key interactions between the inhibitor and the enzyme that supported the importance of the aryltriazole moiety and was in agreement with the improved trafficking of GCase observed in immunolabeling experiments performed by the authors.

#### 2.1.3. Bicyclic Iminosugars

Calystegines are polyhydroxy nortropane alkaloids that display GCase selectivity over other lysosomal enzymes [44]. García-Moreno et al. [45] reported the synthesis and biological evaluation of several amphiphilic glycomimetics presenting a rigid nortropane skeleton based on 1,6-anhydro-L-idonojirimycin, formulated as the corresponding inclusion complexes with β-cyclodextrin (βCD). The insertion of a terminal polyfluorinated fragment allowed the improvement of the complex stability and GCase-rescue capabilities due to additional interactions with the protein surface according to medicinal chemistry principles [46,47], improving membrane crossing, also including the blood–brain barrier (BBB). The authors encapsulated their fluorinated derivatives in form of β-cyclodextrin complexes, taking advantage of the affinity of this kind of compound for the cavity of β-cyclodextrin and avoiding the intrinsic aggregation previously observed for polyfluorinated derivatives [48], thus gaining in drug availability and cell permeability.

The new compounds presented an alkyl or fluorinated chain of different lengths, and all of them showed strong competitive inhibition towards the mammalian β-glucosidase enzymes selectively. The encapsulated βCD complexes (*K*_i_ = 0.09–8.2 µM) behaved as stronger inhibitors than compounds **57**–**60** (*K*_i_ = 0.33–90 µM) (Figure 12) alone (without encapsulation). In the case of GCase, the complexes were better inhibitors at pH 7 than at pH 5, which is a positive feature for a chaperone candidate.

The complexes of **57**–**60** with βCD were tested as chaperones in fibroblasts of Gaucher patients bearing the following mutations: N370S/N370S or N370S/84GG (non-neuronopathic, type 1 GD), V230G/R296X or L444P/P415R (acute neuronopathic, type 2 GD), and N188S/G183W or L444P/L444P (neuronopathic, type 3 GD). The best enhancements observed for each mutation are summarized in Table 1. From the results, it can be concluded that the complex **58**:βCD was the one with the widest activity spectrum (Table 1, entries 1, 2, and 4–6). Further immunolabeling experiments demonstrated that the chaperone-action mechanism involved rescuing and trafficking of GCase, as the recovered enzyme was colocalized in the lysosome and no longer in the ER after treatment of the cells with **58**:βCD. The results on the L444P/L444P lines are remarkable (Table 1, entry 6), since these cell lines are resistant to most PCs.

Pyrrolizidines are another class of bicyclic iminosugars that has been widely studied in the field of glycosidase inhibition [49]. Parmeggiani et al. [29] completed their screening on pyrrolidines and piperidines with several pyrrolizidine iminosugars obtained from nitrones. Unfortunately, the great majority of pyrrolizidines did not impart any relevant inhibition towards GCase, except for compounds **61** and **62**, the latter bearing a phenyl-substituted triazole moiety, thus showing the importance of the incorporation of an aromatic group in the molecule (Figure 13). However, when tested on cell lines, these compounds did not give any enzymatic rescue.

Mena-Barragán et al. [50] prepared a family of bicyclic iminothiazolidine-sp^2^-iminosugars derived from d-fagomine, DAB (**21**, Figure 8) and LAB (***ent*-21**, Figure 9) bearing alkyl chains of different lengths (4, 8, or 16 carbon atoms). The bicyclic compounds **63**–**66**, ***ent*-65**, and ***ent*-66** (Figure 14) were better inhibitors than the corresponding monocyclic counterparts, especially those with a longer alkyl chain (octyl and hexadecyl), and showed IC_50_ values towards GCase in the low micromolar range (IC_50_ = 4.1–12.4 μM). Interestingly, the compounds were one order or magnitude more potent inhibitors at pH 7 than at pH 5.

These compounds were tested as chaperones in fibroblasts of GD patients bearing the N188S/G183W, V230G/R296X or L444P/L444P mutations, which are associated with type 2 or type 3 neuronopathic variants. Unfortunately, only compound **63** was able to increase the GCase activity by 1.3-fold in the N188S/G183W and V230G/R296X fibroblasts at 2 μM. Regarding the homozygous L444P variant, none of the compounds was effective.

Ortiz Mellet and coworkers reported the synthesis of a series of sp^2^-iminosugars with different structural requirements, in order to evaluate the effect of these modifications on the inhibitory activity towards GCase and their chaperoning capabilities in GD cell lines [51]. When tested on a panel of commercial glycosidases and human GCase, the best chaperone candidates were compounds **67**–**70** (Figure 15). The GCase-chaperoning capabilities of these compounds were evaluated in type 3 GD fibroblasts with the N188S/G183W mutation, and the results showed that the non-reducing derivatives behaved as better chaperones than the corresponding reducing analogues (**69** and **70** vs. **67** and **68**, Table 2, entries 3 and 4 vs. 1 and 2). The presence of a sulfur atom in the iminosugars also led to an improvement in chaperone behavior with respect to the oxygenated counterparts (**68** and **70** vs. **67** and **69**, Table 2, entries 2 and 4 vs. 1 and 3). It is worth highlighting that all derivatives showed higher GCase inhibitory activity at neutral pH (Table 2), which is in favor of a good chaperone candidate, as a higher binding affinity in the ER is required to ensure the correct folding of the enzyme. Compounds **67**, **68**, and **69** had previously been tested in other GD cell lines, displaying moderate-to-good chaperoning effects [31,52,53], but the new derivative **70**, which presented the most suitable structure (non-reducing and sulfured-sp^2^-iminosugar), emerged as the most active chaperone candidate reported to date for a neuronopathic Gaucher mutation, as it enhanced GCase activity even in the picomolar range (1.7-fold at 20 pM).

#### 2.1.4. Multivalent Iminosugars

The multivalent display of carbohydrates on adequate platforms can afford more potent glycosidase inhibitors than the corresponding monovalent counterparts. Despite many efforts to develop multivalent glycosidase inhibitors in recent years [54,55,56,57,58,59], this strategy has been proved successful mainly in the field of α-mannosidases [54,55,56,57,58,59,60,61], with few examples of multivalent derivatives targeting the glycosidases implicated in the LSDs [62,63,64]. More recently, some of us have described multivalent derivatives with potential as pharmacological chaperones in fibroblasts of Fabry patients [65]. Regarding GCase and Gaucher disease, the first examples of multivalent PCs were reported by Compain and coworkers 10 years ago. The authors identified the iminosugar–cyclodextrin conjugate **71** [66] (Figure 16) and the tetravalent DNJ derivative **72** [67] (Figure 16) as potent PCs able to maximally increase GCase activity in N370S Gaucher fibroblasts up to 2.4-fold and 3.3-fold at 10 µM, respectively (Table 3, entries 1 and 2) [67]. Furthermore, the trivalent acetyl-DNJ-derivative **74** (Figure 16), although being a worse inhibitor, showed higher enhancement in GCase activity than its corresponding unprotected analogue **73** (Figure 16) (3.0-fold vs. 2.4-fold, Table 3, entries 4 vs. 3) in N370S GD fibroblasts at one order of magnitude lower concentration (1 µM vs. 10 µM, Table 3, entries 4 vs. 3) [67]. The authors argued that compound **74**, with improved permeability and cellular uptake with respect to **73**, may undergo acetyl hydrolysis in vivo, providing the first evidence of the potential of prodrugs in PCT.

Later on, Laigre et al. [68] prepared a set of DNJ-based multivalent clusters with free OH (compounds **72**, Figure 16, and **76**, Figure 17) or in their polyacetylated version (compounds **75** and **77**, Figure 17) in order to better investigate the prodrug effect. They also incorporated a morpholine moiety in some derivatives to study the chaperone effect in Gaucher disease by synthesizing lysosome-targeting probes [69]. The GCase inhibition and enhancement results are summarized in Table 4.

The best enhancement was obtained for **72** (2.5-fold at 1 μM, Table 4, entry 1) although this compound was not the best inhibitor. Unfortunately, no prodrug effect was observed, as all the peracetylated derivatives showed decreased or similar chaperoning activity with respect to the deprotected analogues (**75** vs. **72** and **77** vs. **76**, Table 4, entries 2 vs. 1 and 4 vs. 3). In addition, the insertion of the morpholine moiety did not further improve GCase-activity enhancement (**76** vs. **72** and **77** vs. **75**, Table 4, entries 3 vs. 1, and 4 vs. 2). Since the replacement of an iminosugar unit (both deprotected and peracetylated) with a morpholine moiety resulted in comparable chaperoning activity, it can be argued that the morpholine moiety acts as an iminosugar surrogate, rather than a lysosome-targeting probe.

Stauffert et al. [70] developed a library of dimers derived from DIX (**2**, Figure 2) and its enantiomer with the aim of simultaneously targeting the active site and a secondary binding site of GCase. The library was composed of five heterodimers (compounds **78a**–**e**) and two homodimers (compounds **79** and ***ent*-79**) of both enantiomers (Figure 18). The corresponding monovalent counterparts (**80**–**83** and ***ent*-80** and ***ent*-83**) were also prepared and evaluated towards GCase (Figure 18 and Table 5).

All compounds resulted to be GCase inhibitors in the micromolar or nanomolar range, and some of them exhibited a better inhibition at pH 7, which means a higher affinity towards GCase at the neutral pH of the ER. With the exception of compounds **80** (Table 5, entry 8) and **82** (Table 5, entry 10), all the new derivatives were non-competitive inhibitors. The non-competitive inhibition mode observed for these compounds indicated that they were able to bind GCase in a secondary binding site. The best inhibition results at pH 7 were obtained for monomers **80**, **82**, and **83** (Table 5, entries 8, 10 and 11) and dimers **78e** and **79** (Table 5, entries 5 and 6). All compounds were tested as GCase enhancers in a panel of fibroblasts from GD patients with three different genotypes (N370S/N370S, L444P/L444P or G202R/G202R), and good results were observed only with G202R/G202R fibroblasts. In the monomeric series, the best performances were obtained for compounds **80**, **82**, and **83** (Table 5, entries 8, 10 and 11), **82** (3.6-fold at 100 nm, Table 5, entry 10) being the best enhancer, even if this compound is not the most potent inhibitor of the series. Among the dimeric analogues, heterodimer **78e** and homodimer **79** gave the most outstanding results (2.2-fold and 2.5-fold at 100 nM, respectively, Table 5, entries 5 and 6), but they did not surpass the enhancement observed for the corresponding monomeric counterpart **82** (Table 5, entry 10). Decreased glucosylceramide levels and colocalization studies further confirmed the observed results. It is worth highlighting that the absolute configuration of the iminosugar is crucial for GCase enhancement, as dimers **78e** and **79** showed similar enhancements, while dimer ***ent*-79** was inactive (Table 5, entries 5–7).

Very recently, Cardona, Matassini and coworkers reported the synthesis of five new multivalent iminosugars, obtained through CuAAC reaction of an azido ending trihydroxypiperidine with several propargylated scaffolds (Figure 19), designed to investigate how different valences and topologies affect the GCase response [71].

The whole set of inhibitors showed GCase inhibition in the low micromolar range, with lowest IC_50_ values obtained for the tri- and hexavalent compounds **85** and **87** (IC_50_ = 7 µM and 6 µM, respectively). While negligible chaperoning activity was found for compounds **85**, **86**, and **88** in fibroblasts from GD patients bearing the N370S/RecNcil mutations (1.2- to 1.3-fold at 10–50 µM), a remarkable 2-fold activity enhancement at 10 µM was obtained for the trivalent **85** in GD patients’ fibroblasts bearing the homozygous L444P mutation. This enhancement is one of the highest ever observed for a PC towards these PC refractory cell lines.

### 2.2. Sugar Analogues

Apart from iminosugars, other carbohydrate-derived analogues have been studied as possible chaperones for the treatment of GD. Castilla et al. [72] reported the synthesis of pyranoid-type glycomimetics with a *cis*-1,2-fused glucopyranose-2-alkylsulfanyl-1,3-oxazoline (Glc-PSO) structure. Based on previous results [73], which suggested that the nature of the moiety attached to the sulfur atom highly influenced the inhibitory potency against human GCase, the authors investigated the inhibitory properties of compounds incorporating functional groups of different nature (fluorine, iodine, methyl ester, and carboxylic acid, compounds **89**–**92**, Figure 20). These compounds can be considered conformationally locked *N*-glycosides, due to their bicyclic fused skeleton formed by a six-membered and a five-membered cycle, which imposes a skew-boat conformation on the pyranose ring. This conformation was found to be responsible for the bioactivity, since it imparts to the compounds a transition state mimic feature [74].

The results showed a clear influence of the terminal group of the chain on inhibitory activity, with compound **89** incorporating a fluorine atom being the most active derivative with an IC_50_ value of 3.9 μM. They also observed an increase in IC_50_ value when the pH was changed from 7 to 5, which is in favor of a good chaperone candidate. The observed variability agrees with the initial hypothesis of the additional stabilization of the enzyme–inhibitor complex, due to interactions of this terminal group with the enzyme. The most active compounds, **89** and **91**, were evaluated as chaperones in fibroblasts of patients with the N370S/N370S or L444P/L444P mutations. The best results were obtained for compound **89** towards N370S/N370S-mutated fibroblasts, detecting an improvement of 62% at 30 μM, which was higher than that observed for ambroxol (see Section 2.3) at the same concentration, probably due to the overcoming of the inhibitory effect. Unfortunately, no improvement was observed in fibroblasts bearing the L444P/L444P mutation.

Navo et al. [75] developed a family of 3-amino-6,7-dihydroxy-3,5-bis(hydroxymethyl)hexahydropyrano[3,2-*b*]pyrrol-2(1*H*)-one derivatives (APP) obtained from tri-*O*-benzyl-2-nitro-d-galactal following a methodology previously reported by the authors [76]. The new derivatives incorporated a palmitoyl fragment to evaluate the effect of this hydrophobic moiety on the inhibitory profile. After a screening towards a panel of commercial glycosidases, derivatives **93**–**96** (Figure 21) were the most promising candidates among all the compounds prepared and were subsequently tested towards human GCase.

From the results obtained, it was deduced that the presence of an *O*-acyl substituent in the aglycone was detrimental to inhibitory activity, as compound **94** (*K*_i_ = 20 μM at pH 7) was a 1.5 to 2.5-fold more potent inhibitor than derivatives **93**, **95**, and **96** (*K*_i_ = 33–52 μM at pH 7). The authors also observed that compounds **93**–**96** behaved as stronger inhibitors of GCase at pH 7 than at pH 5. These compounds were assayed as chaperones in fibroblasts of GD patients with the N370S/N370S, F213I/F213I, or L444P/L444P mutations, associated with type 1, 2, and 3 GD, respectively. Only compound **94** displayed a significant enhancement in GCase activity at 20 μM for the N370S/N370S and F213I/F213I mutations (1.3- and 1.5-fold increase, respectively).

To justify the better biological properties of compound **94** over derivatives **93**, **95**, and **96**, molecular dynamics (MD) simulations were performed and the calculations showed that the pyranoid glycone moiety established hydrogen bonding in a similar way to other competitive inhibitors of the glycomimetic family. The amide moiety also participated in the formation of hydrogen bonds with the enzyme, thus influencing the orientation of the aliphatic palmitoyl chain towards the hydrophobic region that gives access to the active site of the enzyme. This shift allowed the gain of additional interactions that reinforced interaction with the GCase enzyme.

### 2.3. Miscellaneous Examples

The use of activity-based probes (ABPs) can be helpful to understand the stabilization mechanism of GCase through inhibitor binding. Ben Bdira et al. [77] described the use of cyclophellitol-derived ABPs that bind the GCase active site irreversibly to the catalytic nucleophile E340. The amphiphilic ABPs studied in vitro and in situ displayed a fluorescent hydrophobic green or red BODIPY tail (compounds **97** and **98**, Figure 22). These compounds were able to enhance thermal stability of GCase and sensitivity to tryptic digestion. The findings pointed out that ABP labeling stabilizes GCase against proteolytic degradation in the lysosomes and did not avoid *N*-glycan modifications by lysosomal glycosidases. These types of compounds can be valuable marker tools to study the contribution of hydrophobic interactions to GCase stabilization by selective labeling of the active form of GCase in living cells and laboratory animals.

In 2009, Maegawa et al. screened a library of 1040 FDA-approved drugs through a thermal denaturation assay using wild-type GCase, and identified ambroxol (ABX, **99**, Figure 23), a cough medicine, as a pH-dependent, mixed-type inhibitor of GCase [78]. In particular, ABX (**99**) showed its maximal inhibitory activity at neutral pH (8.1 µM), but turned out to be a negligible inhibitor of GCase at acidic pH (31 µM) that can be found in the lysosomes, a crucial tandem of properties for a good chaperone candidate [79].

In vitro screening of ABX demonstrated low cytotoxicity (60 µM in GD patients’ fibroblasts) and good chaperoning activity with F213I/F213I, R120W/L444P, N188S/G193W, N370S/N370S, and F213I/L444P mutant cells, while no effects were observed with the L444P/L444P mutant cells or the control cell line. The chaperoning activity and toxicity of ABX (**99**) were also tested in a wild-type mouse model, showing increased GCase activity in some tissues (namely, cerebellum, heart, and spleen) and demonstrating that ABX (**99**) penetrates the BBB, exerting its activity as PC in the brain [80,81]. However, after a pilot study was carried out in 2013 to assess the tolerability and efficacy of ABX (**99**) as a PC in patients with symptomatic type 1 GD [82], this PC still appears stuck in phase II of the clinical trial [32]. In addition, ABX (**99**) is currently being evaluated in a phase II trial involving 75 patients with moderate PD [83], and preliminary data indicate good brain distribution and safety profile [84].

## 3. Allosteric Enhancers

The use of inhibitory chaperones can be quite challenging, considering that GCase needs to overcome the inhibitory effect in the lysosomes once it is properly folded in order to carry out its biological action. As such, the search for new PCs that do not bind the active site of the enzyme is an emerging alternative that has gained interest in the last few years. In particular, since ambroxol was first identified as a chaperone for GCase through high-throughput screening (HTS), this technique has turned out to be useful for the search of other non-carbohydrate-derived molecules with potential as allosteric binders of GCase [85].

Compounds NCGC607 (**100**) and NCGC758 (**101**) were discovered in a second HTS of a library of small molecules composed of 250,000 compounds (Figure 24) [86]. Aflaki et al. [87] studied the treatment of the cells of five patients with GD type 1 or 2 with or without PD with the noninhibitory salicylic acid derivative NCGC607 (**100**). The cells showed restored GCase activity and protein levels, reduced glycolipid storage, and reduced α-synuclein accumulation in patients with PD. These results also suggested that compound NCGC607 (**100**) chaperoned mutations other than N370S, as enhanced activity was observed in patients with type 2 GD with the L444P mutation. Compound NCGC758 (**101**), with a pyrazolopyrimidine moiety in its structure, also demonstrated beneficial properties for the treatment of GD.

Sun and Wang reported a screening of noninhibitory compounds and found that the racemic mixture **102** (Figure 24) behaved as a good enhancer (ca 8-fold at 10 μM) in mouse fibroblasts bearing the V394L/V394L mutation [88].

In this context, Zheng et al. identified a series of extremely potent non-iminosugar GCase modulators through a systematic structure–activity relationship (SAR) study on quinazoline derivatives. In particular, compound JZ-4109 (**103**, Figure 24) was able to significantly increase GCase protein abundance and GCase activity in homozygous N370S-mutant fibroblast cells from a type 1 GD patient (about 2-fold at 2 µM) [89]. Even more interestingly, the authors characterized (by native mass spectrometry and X-ray crystallography) an allosteric binding site for these modulators by conducting a covalent modification of a lysine residue close to a properly modified ligand. Through transmission electron microscopy (TEM) experiments, they also provided insight into the mechanism of action of **103**, suggesting that GCase stabilization is due to the ability of the ligand to induce GCase dimerization [90].

A further confirmation of the role played by dimerization in GCase activation was recently provided by Rodríguez Sarmiento and coworkers. Indeed, the authors observed through analytical ultracentrifugation (AUC) that GCase dimerization was induced by the pyrrolo[2,3-*b*]pyrazine **104** (Figure 24), identified by testing on a cellular assay 400 diverse heterocyclic compounds, selected according to the literature and filtered by molecular properties. The GCase activator that emerged from this study did not bind to the GCase catalytic site, but to a new pocket at a dimer interface, as supported by X-ray and surface plasmon resonance (SPR) competition experiments. These results gave important mechanistic insight on the field of GCase activation, paving the way to the development of a new generation of GCase activators with CNS drug-like properties, for the treatment of Gaucher and Parkinson’s disease [91].

Other examples of allosteric GCase binders were provided by García Collazo et al. The authors reported an extensive biological screening of non-carbohydrate-derived compounds that were allosteric binders of mutated GCase. Chaperone assays on mutated homozygous L444P human fibroblasts showed moderate enhancements in the 6–50 μM concentration range. Among all compounds tested, the better enhancements (>2-fold) were observed for the synthetic compounds **105**–**108** and the commercially available derivative **109** (Figure 25) [92].

## 4. Studies in Parkinson’s Animal Models

Parkinson disease (PD) is the second most common neurodegenerative disorder affecting approximately 1% of the population over 60 and 5% over 80–85 [93]. The pathological hallmarks of PD are formation of α-synuclein (α-syn) aggregates and marked loss of dopaminergic neurons in the *substantia nigra pars compacta* [94]. GCase mutations were shown to be a major risk factor of developing PD: approximately 7–12% of PD patients carry *GBA* mutations [95], and 25% of GD patients have a first- or second-degree relative with PD [96].

In addition, it emerged that enhancing GCase, also in the absence of GD mutations, may represent a valid therapeutic strategy for sporadic forms of PD [97]. The exact mechanism through which GCase deficiency contributes to PD pathogenesis is still unclear, but it includes the accumulation of α-syn, impaired lysosomal function, and ER-associated stress. Impairment of the autophagy–lysosomal pathway plays a principal role in PD pathogenesis and in particular its effect on abnormal accumulation of α-syn with formation of oligomers and fibrils [98]. The relevant examples of GCase chaperones in PD animal models are summarized in Table 6.

Several drug screenings have identified a number of candidates, including already known drugs (drug-repositioning or drug-repurposing strategy), such as ambroxol (ABX, **99**, Figure 23), a drug used in the treatment of respiratory diseases, and IFG (**1**, Figure 2), under clinical phase trials for the treatment of Gaucher disease. The fruit fly *Drosophila melanogaster* shows that development of PD in carriers of GD mutations results from the presence of mutant *GBA* alleles. Expression of mutated *GBA* in *Drosophila* resulted in dopaminergic neuronal loss, a progressive locomotor defect, and increased levels of ER stress. In a *Drosophila* model, both ABX (**99**) and IFG (**1**) have been shown to effectively reduce ER stress and the locomotor deficit, as reported by Schapira and Horowitz and their coworkers [99,100].

Moreover, ABX (**99**) treatment resulted in increased brain GCase activity in a transgenic mouse model expressing the heterozygous L444P mutation in the murine *GBA* gene, a PD model of healthy carrier of GD (*GBA*-PD model) [101].

Conversely, several studies have been performed on transgenic mice overexpressing human wild-type α-synuclein (Thy1-αSyn), which reproduced key features of Parkinson’s disease in the absence of GCase mutations, including nonmotor deficits, pathological, biochemical, and molecular alterations, and at older ages, alterations in nigrostriatal dopaminergic neurons, including loss of striatal dopamine (PD models) [101,102]. Oral treatment with ABX (**99**) and IFG (**1**) increased wild-type GCase activity in the brain of Thy1-αSyn mice, inducing a decrease of α-syn accumulation in dopaminergic neurons of the substantia nigra and improving motor and nonmotor functions. The pharmacodynamic properties of these compounds in the brain of mouse models demonstrated their ability to cross the blood–brain barrier (BBB) [101,102]. Moreover, Migdalska-Richards et al. also reported the ability of ABX (**99**) to increase brain GCase activity of 20% in the midbrain of wild-type nonhuman primates after oral chronic treatment, demonstrating, once again, that ABX (**99**) crosses the blood–brain barrier (BBB) [103].

Finally, Burbulla et al. [104] developed compound S-181 (**110**, Table 6), a quinazoline-derived modulator of GCase with good pharmacokinetic properties. Intraperitoneal single administration of S-181 into wild-type mice showed a maximal concentration of 2.9 µM in brain and a half-life of 20.7 h, with higher concentrations in brain than in plasma after 2 h. Compound S-181 (**110**) enhanced wild-type GCase activity in heterozygous D409V mice, a *GBA*-PD model, resulting in reduction of the lipid substrates and α-syn in brain.

In consideration of the preliminary results concerning the ability to cross the BBB, the compounds described in Table 6 represent promising therapeutics for future development.

## 5. Conclusions

The human lysosomal enzyme acid-β-glucosidase (also known as glucocerebrosidase or GCase) has recently become an increasingly interesting biological target to address not only the most common form of lysosomal storage disorders (LSDs), Gaucher disease (GD), but also much more common neurological pathologies, such as Parkinson’s disease (PD). In this regard, pharmacological chaperone therapy (PCT), which employs small compounds able to bind and stabilize the misfolded lysosomal enzymes, facilitating their translocations to the lysosomes, and resulting in a net recovery of the compromised catalytic activity, has recently emerged and been placed on the market of the first PC for an LSD (e.g., Galafold, for the treatment of Fabry disease). However, there is no PC-based drug for the treatment of GD or PD, yet. Iminosugar-based glycomimetics are among the most investigated class of PC for GD. They are lysosomal GCase inhibitors that act as a template, stabilizing the native folding state of the enzyme in the endoplasmic reticulum (ER), with a consequent final counterintuitive effect of enhancing the lysosomal enzyme catalytic activity in the lysosomes.

Alkylated iminosugars have great potential as PCs for the GCase enzyme. It emerged that the position and length of the alkyl chain is crucial to impart the desired activity, which is not possible to be determined a priori with docking studies, since during in vitro tests with mutated cell lines, other important factors take place, such as membrane permeability and cell viability. Overall, an octyl alkyl chain seems to represent the optimal compromise between good affinity with the enzyme active site and good cell permeability, with negligible toxicity. Iminosugars of different types were identified recently as good PCs, also towards mutated cell lines (e.g., L444P, present in neurological forms of the disease), which were quite resistant to the majority of the PCs found in the past. The in vitro enzymatic enhancement observed with iminosugar-based PCs is usually around 2-fold, which is reached with different concentration of compounds ranging from micromolar to nanomolar. However, this aspect strongly depends on compound toxicity and cell permeability, so it is difficult to predict a general trend. Nevertheless, chaperoning activity observed at nanomolar concentration is in principle better.

In most of the reports, the selectivity towards GCase over other lysosomal glycosidases was also verified, which is important to avoid undesired side effects.

The main limitation of the competitive inhibitor-based PCs is that for their application in vivo, careful dosage tuning is needed to overcome the inhibitory effect. For this reason, more recently, allosteric enhancers have been identified through HTS screening or drug repurposing. This review aimed to describe the efforts made by the scientific community in the last 7 years (since 2015) in order to identify new PCs for the GCase enzyme. We deem that the identification of an effective PC for GCase is of fundamental importance not only for Gaucher disease but also for other diseases related to the misfolding of this enzyme, such as Parkinson’s disease, and we are confident that the scientific community will find solutions based on this therapy to be placed on the market.

## Figures and Tables

**Figure 1 pharmaceuticals-15-00823-f001:**
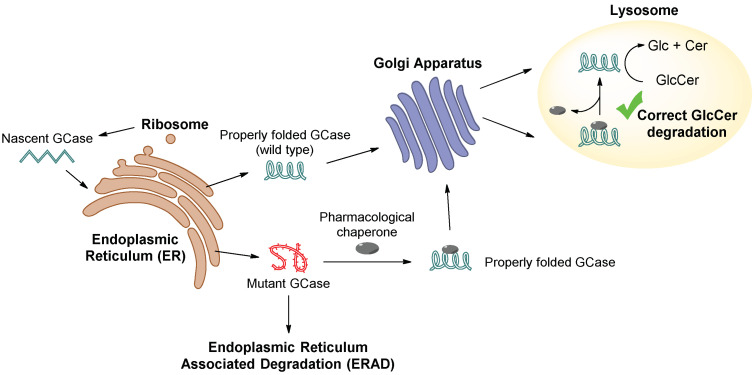
Schematic representation of the action of a pharmacological chaperone (PC) on mutant GCase, which recovers a correct GlcCer degradation.

**Figure 2 pharmaceuticals-15-00823-f002:**
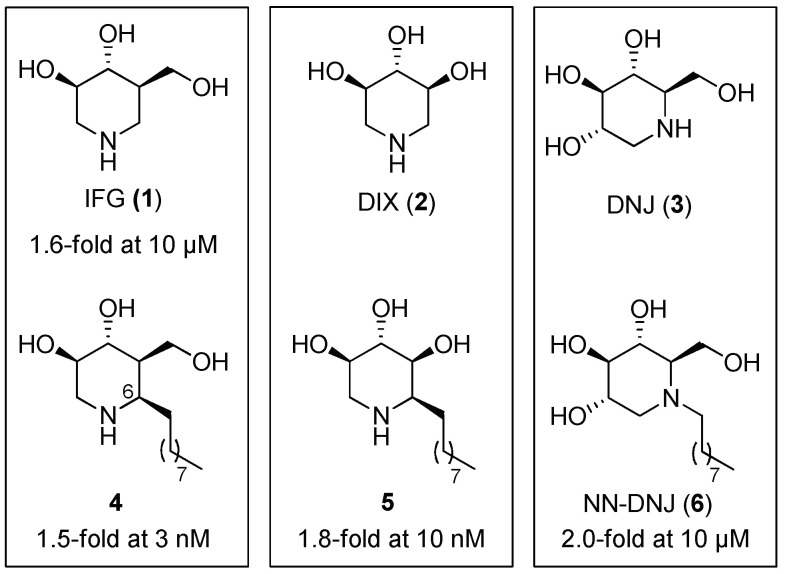
GCase activity increase observed in N370S GD fibroblasts in the presence of IFG (**1**), DIX (**2**), and DNJ (**3**) and their alkylated derivatives (compounds **4**–**6**).

**Figure 3 pharmaceuticals-15-00823-f003:**
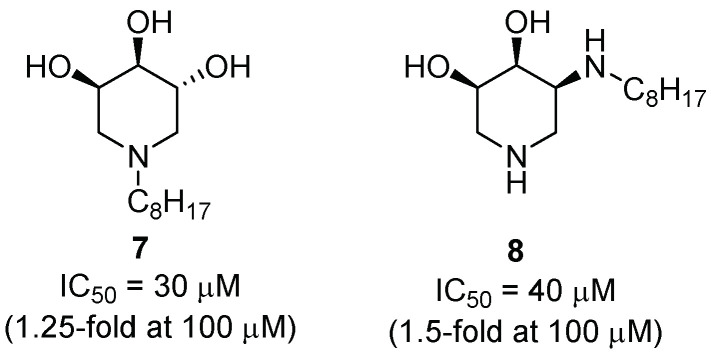
Inhibitory activity and GCase enhancement of alkylated glycomimetics **7** and **8**.

**Figure 4 pharmaceuticals-15-00823-f004:**
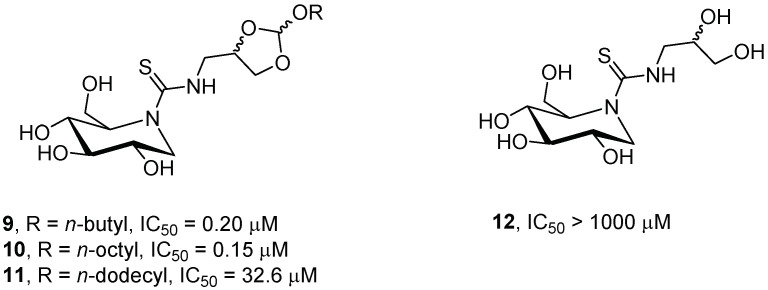
GCase inhibitory activity of DNJ-based sp^2^-iminosugars **9**–**12**.

**Figure 5 pharmaceuticals-15-00823-f005:**
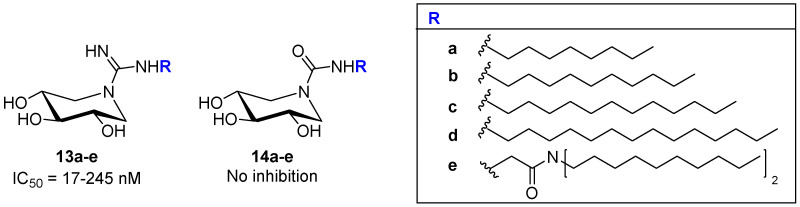
GCase inhibitory activity of DIX–derived guanidine and urea compounds **13** and **14**.

**Figure 6 pharmaceuticals-15-00823-f006:**
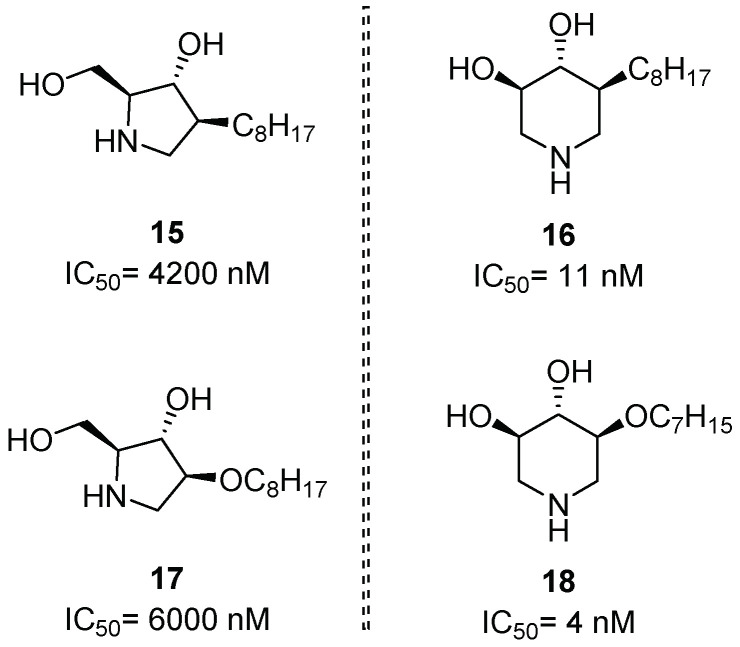
GCase inhibitory activity of alkylated pyrrolidines **15** and **17** and their piperidine analogues **16** and **18** obtained through ring expansion.

**Figure 7 pharmaceuticals-15-00823-f007:**
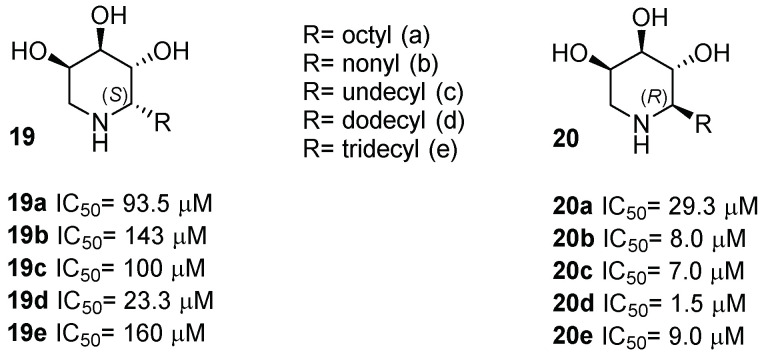
GCase inhibitory activity of the trihydroxypiperidine glycomimetics **19** and **20**.

**Figure 8 pharmaceuticals-15-00823-f008:**
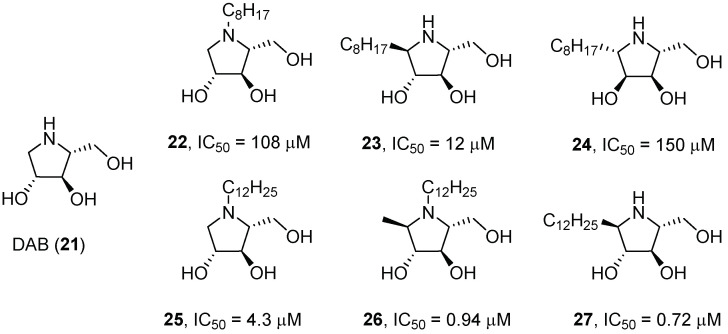
*N*- and *C*-alkylated pyrrolidine derivatives analogues of DAB (**1**) as GCase inhibitors.

**Figure 9 pharmaceuticals-15-00823-f009:**
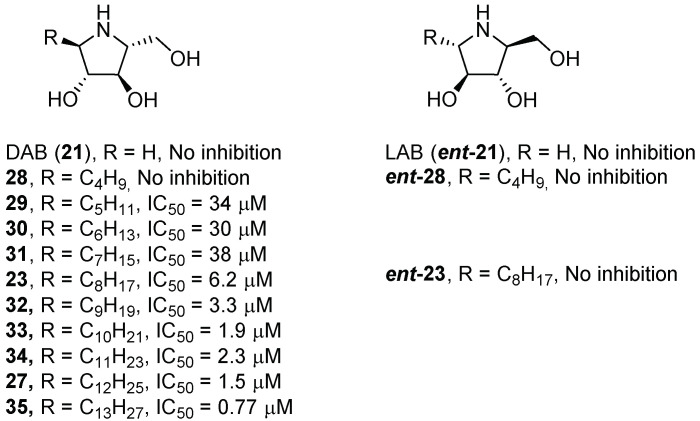
DAB and LAB alkylated analogues and their activity towards GCase.

**Figure 10 pharmaceuticals-15-00823-f010:**
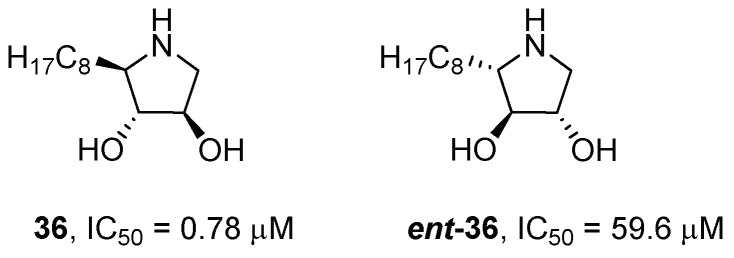
Compounds **36** and ***ent*-36** lacking the hydroxymethyl moiety of DAB and LAB.

**Figure 11 pharmaceuticals-15-00823-f011:**
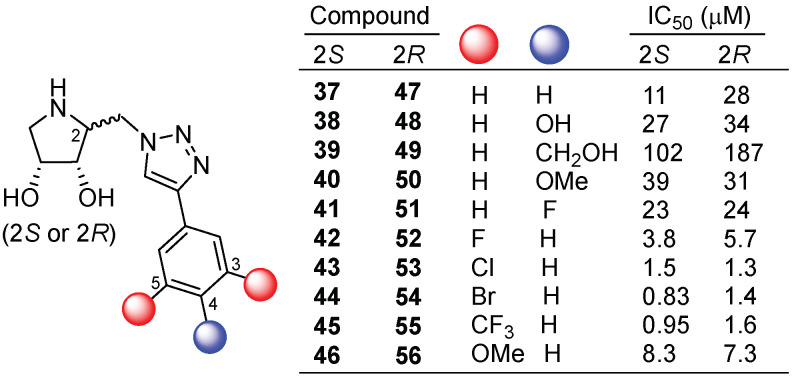
Dihydroxypyrrolidines **37**–**56** decorated with an aryltriazole moiety.

**Figure 12 pharmaceuticals-15-00823-f012:**
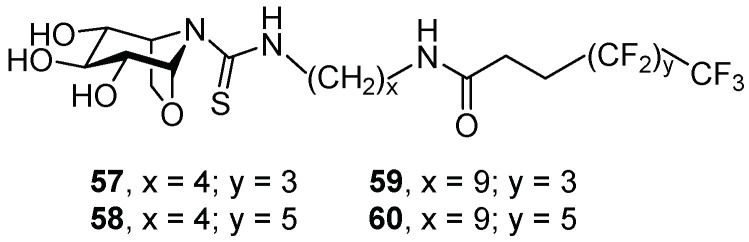
Polyfluorinated nortropanes **57**–**60**.

**Figure 13 pharmaceuticals-15-00823-f013:**
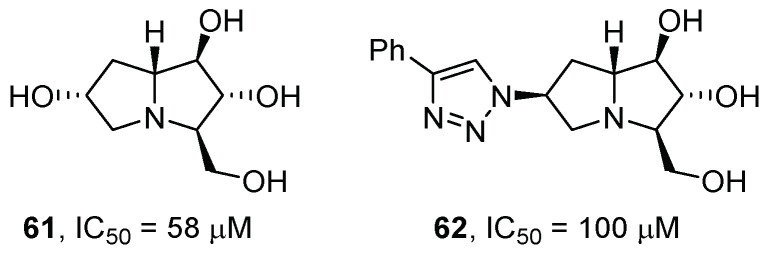
Pyrrolizidine iminosugars **61** and **62**.

**Figure 14 pharmaceuticals-15-00823-f014:**
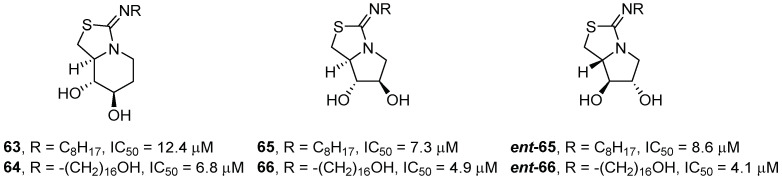
Bicyclic iminothiazolidine-sp^2^-iminosugars.

**Figure 15 pharmaceuticals-15-00823-f015:**
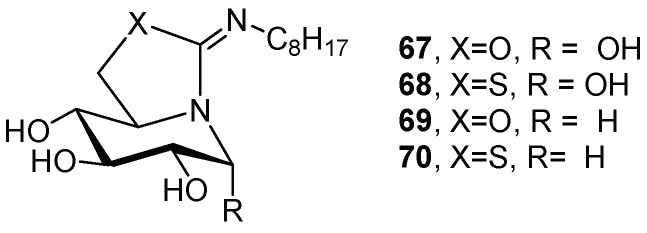
sp^2^-Iminosugars **67**–**70**.

**Figure 16 pharmaceuticals-15-00823-f016:**
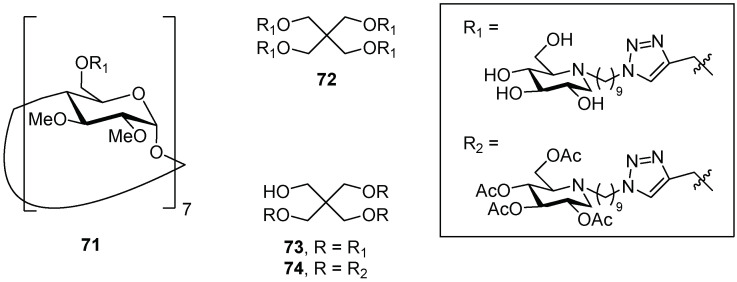
Multivalent iminosugars **71**–**74**.

**Figure 17 pharmaceuticals-15-00823-f017:**
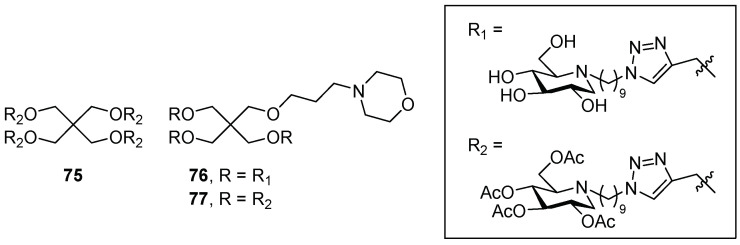
Multivalent iminosugars **75**–**77**.

**Figure 18 pharmaceuticals-15-00823-f018:**
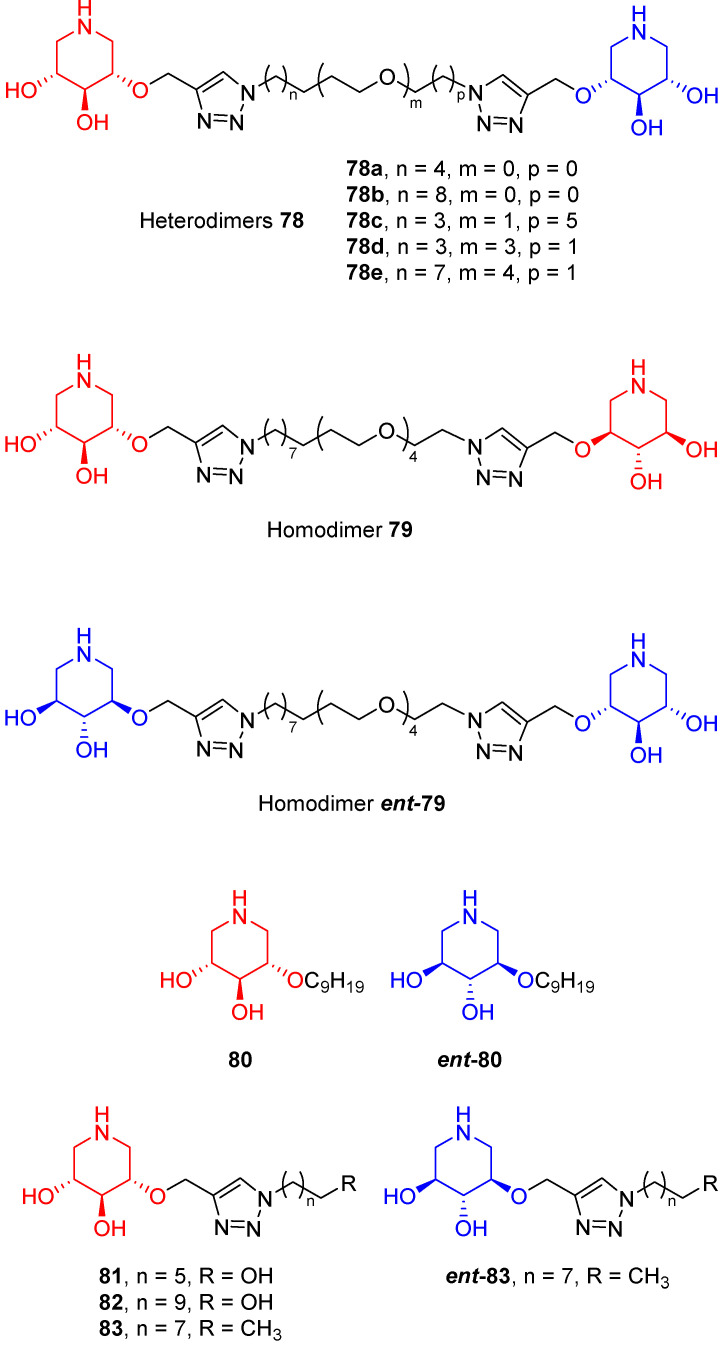
Homo- and heterodimers derived from DIX (**2**) and its enantiomer.

**Figure 19 pharmaceuticals-15-00823-f019:**
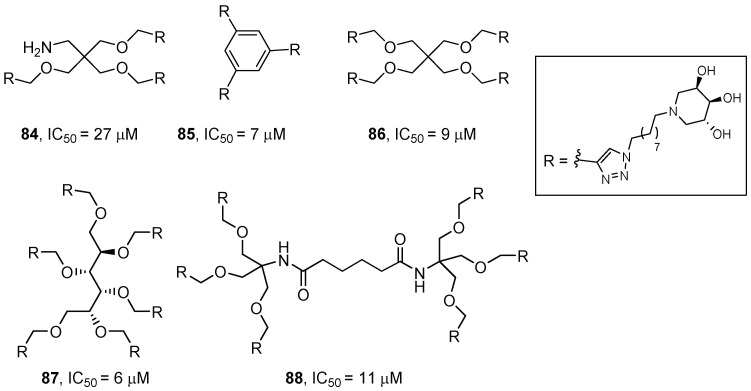
Multivalent iminosugars **84**–**88**.

**Figure 20 pharmaceuticals-15-00823-f020:**
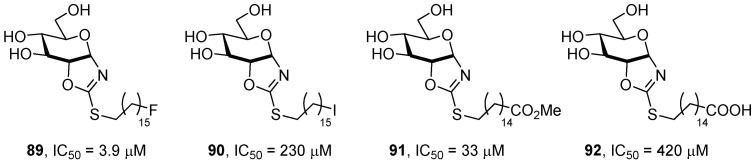
Pyranoid-type glycomimetics **89**–**92**.

**Figure 21 pharmaceuticals-15-00823-f021:**
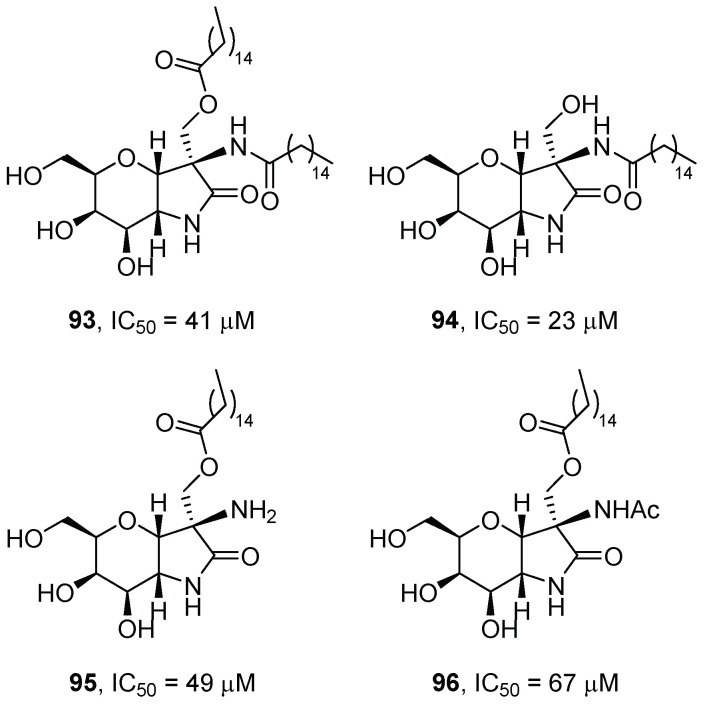
Pyranoid-type glycomimetics **93**–**96**.

**Figure 22 pharmaceuticals-15-00823-f022:**
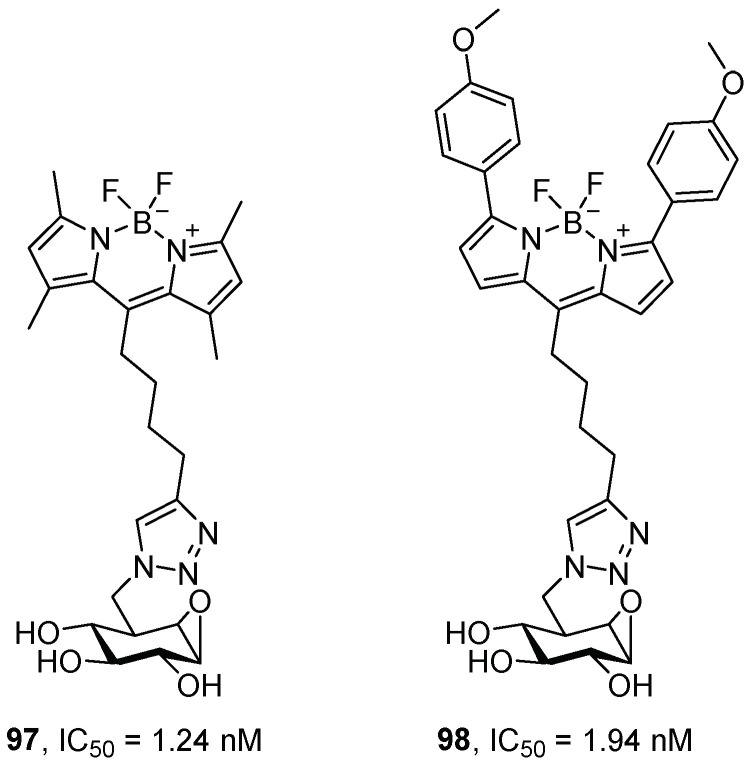
Fluorescent cyclophellitol derivatives **97** and **98**.

**Figure 23 pharmaceuticals-15-00823-f023:**
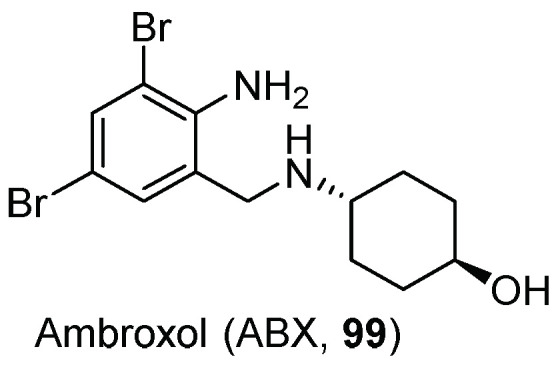
Ambroxol (ABX, **99**), a mixed-type GCase inhibitor.

**Figure 24 pharmaceuticals-15-00823-f024:**
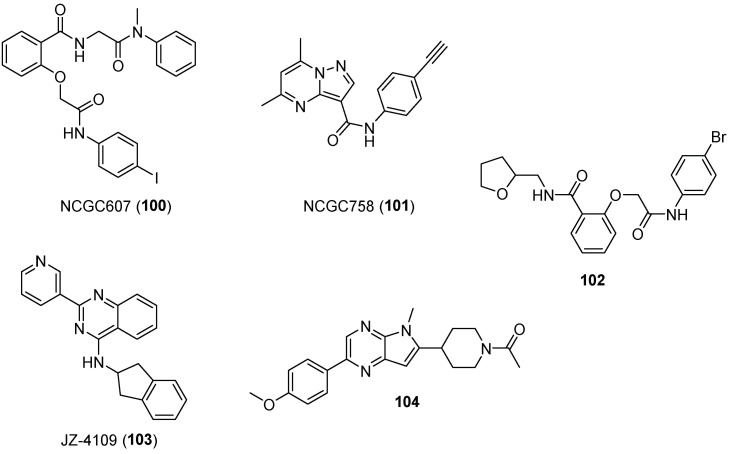
Non-carbohydrate-derived allosteric enhancers **100**–**104**.

**Figure 25 pharmaceuticals-15-00823-f025:**
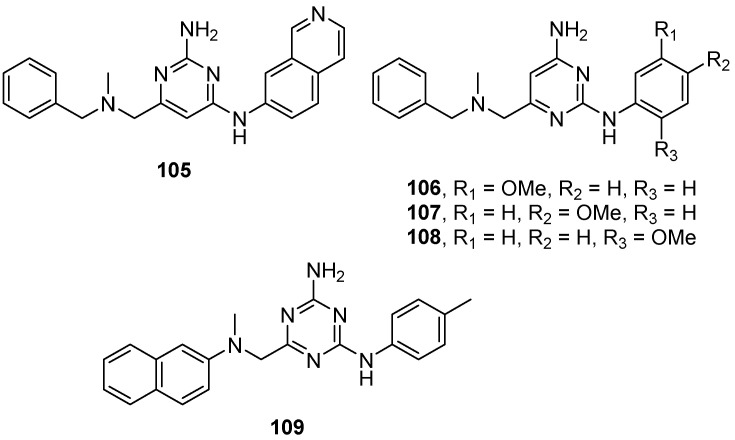
Non-carbohydrate derived GCase modulators **105**–**109**.

**Table 1 pharmaceuticals-15-00823-t001:** Enhancements observed for each Gaucher disease variant.

Entry	GCase Variant	Severity	Best Enhancement	Concentration	Complex
1	N370S/N370S	Type I	1.5-fold	2 μM	**58**:βCD
2	N370S/84GG		2.8-fold	20 μM	**58**:βCD
3	V230G/R296X	Type II	2.7-fold	20 μM	**57**:βCD
4	L444P/P415R		2.9-fold	20 μM	**58**:βCD
5	N188S/G183W	Type III	1.8-fold	20 μM	**57**:βCD and **58**:βCD
6	L444P/L444P		2.8-fold	20 μM	**58**:βCD

**Table 2 pharmaceuticals-15-00823-t002:** *K*_i_ (in µM) and GCase enhancement for compounds **67**–**70**.

Entry	Compound	*K*_i_ (µM) ^a^		Enhancement (N188S/G183W)
		pH 7	pH 5	
1	**67**	15.1	54.4	3-fold at 20 µM
2	**68**	0.26	1.05	3.2-fold at 2 µM
3	**69**	1.7	6.3	3.2-fold at 200 nM
4	**70**	0.013	0.059	3-fold at 2 nM

^a^ Inhibition was competitive in all cases.

**Table 3 pharmaceuticals-15-00823-t003:** GCase enhancement shown by compounds **71**–**74** in N370S Gaucher fibroblasts.

Entry	Compound	Enhancement
1	**71**	2.4-fold at 10 µM
2	**72**	3.3-fold at 10 µM
3	**73**	2.4-fold at 10 µM
4	**74**	3.0-fold at 1 µM

**Table 4 pharmaceuticals-15-00823-t004:** IC_50_ values (in nM) and enhancements observed in N370S Gaucher fibroblasts (at 1 μM).

Entry	Compound	IC_50_ (nM)	Enhancement
1	**72**	500	2.5-fold
2	**75**	n.i.	1.8-fold
3	**76**	109	2.0-fold
4	**77**	n.i.	2.1-fold

n.i. = No inhibition detected at 0.1 mM.

**Table 5 pharmaceuticals-15-00823-t005:** IC_50_ values (in nM) and enhancements observed in G202R/G202R fibroblasts.

Entry	Compound	Valence	IC_50_ (nM)		Enhancement	Concentration (nM)
			pH 5.2	pH 7.0		
1	**78a**	2	1800	223	1.5-fold	2000
2	**78b**	2	23.6	32.5	1.7-fold	2000
3	**78c**	2	57.1	28.5	2.2-fold	2000
4	**78d**	2	1100	362	1.6-fold	2000
5	**78e**	2	46.6	17.0	2.2-fold	100
6	**79**	2	3.6	3.3	2.5-fold	100
7	***ent*-79**	2	162,000	94,000	-	-
8	**80 ^a^**	1	4.3	5.7	1.8-fold	1
9	**81**	1	292	182	1.6-fold	2000
10	**82 ^a^**	1	24.3	19.9	3.6-fold	100
11	**83**	1	4.8	4.3	2.2-fold	10
12	***ent*-80**	1	75,000	42,000	-	300
13	***ent*-83**	1	58,000	19,000	-	2000

All compounds behave as non-competitive inhibitors, unless otherwise noted. ^a^ Competitive inhibitors. “-“: No enhancement observed.

**Table 6 pharmaceuticals-15-00823-t006:** Relevant examples of GCase chaperones in animal models bearing heterozygous *GBA* mutations, developing PD (*GBA*-PD) and in PD animal models.

Compound	Disease Models	Animal Model	Effect of Chaperone	Refs.
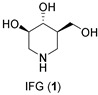	*GBA*-PD	*Drosophila*with mutant *GBA*(N370S and L444P)	- reduce ER stress- reverse locomotor deficits	[99]
PD	Transgenic miceoverexpressinghuman WT α-syn	- improved motor function- reduced α-syn immunoreactivity- reduced α-syn aggregates	[102]
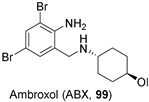	*GBA*-PD	*Drosophila*with mutant *GBA*(N370S and L444P)	- reduce ER stress- reverse locomotor deficits	[99,100]
*GBA*-PD	Gba1^L444P/+^ mice	- increased GCase activity in brains	[101]
PD	Transgenic miceoverexpressinghuman WT α-syn	- reduced α-syn levels	[101]
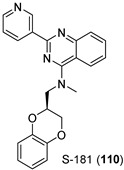	*GBA*-PD	Gba1^D409V/+^ mice	- reduced GlcCer levels in brain- reduced α-syn levels in brain	[104]

## Data Availability

Data sharing not applicable.

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
