# Peer review of "GCase Enhancers: A Potential Therapeutic Option for Gaucher Disease and Other Neurological Disorders"

_pharmaceuticals, 2022, doi:10.3390/ph15070823_

Round 1

Reviewer 1 Report

The review by Cardona and co-workers discuss the potential of pharmacological chaperones (PCs) and enzymatic enhancers in the treatment of lysosomal and Parkinson disease. The review focuses on describing the biological activity of iminosugar-type compounds and non-carbohydrate activity enhancers. This research topic has been extensively investigated during the last decades but was poorly rewarded in terms of marketed drugs. The review is enjoyable to read and may be published in Pharmaceuticals after the following modifications/corrections.

-A significant number of previous reviews focus on PC to treat lysosomal disorders, please specify the originality, different perspective provided by the present work.

-An introductive figure presenting the maturation of glycolipids in the relevant cell compartments with the GCase activity(ER, Golgi, lysosomes) and the chaperoning activity of synthetic molecules may help the reader to better understand the general concept.

-It seems that the in vitro enzymatic enhancement observed with iminosugar type PC is generally limited to 2-fold. In the review several examples of compounds giving this level of enhanced activity but at very different concentrations are provided. It is not clear for the reader at which concentration (micromolar, nanomolar?) a compound is considered as a good pharmacological chaperone in vitro. Please comment if possible.

-In the same vein, target selectivity is crucial when designing new iminosugars to limit detrimental side-effects. However, no indications of glycosidase selectivity is provided in the review.

-Structure-activity relationship is also poorly discussed. The GCase seems to accommodate a high diversity of iminosugars regardless of their cycle size and configuration of hydroxyl groups, making a rational design difficult to achieve. Presenting published structures of GCase-inhibitors, if any, may enrich the review.

- The author presents the chaperoning activity of GCase with iminosugar-based compounds and an allosteric enhancement with non-carbohydrate derivatives. The situation may actually be more complex as these two types of carbohydrate mimics have been previously shown to be able to enhance the enzymatic activity of glycosidases (see:  Carbohydr.Res.2008, 343,2878–2886.) It may be of interest to more generally discuss the faculty of iminosugars to enhance the enzymatic activity in addition of their chaperoning effect.

-Reference 43 is actually not the first example of pharmacophores linked to iminosugars by CuAAc for GCase chaperoning, please see J. Org. Chem. 2011, 76, 19, 7757–7768.

-In the introduction to multivalent structures, rather than citing a mix of reviews and selected articles (ref 54-58), I would rather suggest to only cite all the relevant reviews published in the field (refs are missing).

-Ref 59 “chemistry” should read “chemistry a European journal”

-Ref 62, is it really a significant multivalent effect in chaperoning evidenced in this study? If not the text should be pondered.

Line 511 ‘higher binding affinity for the catalytic pocket’ These compounds are suicidal inhibitors and it should be stated that the IC50 value observed is therefore not comparable with more conventional inhibitors.

Author Response

The review by Cardona and co-workers discuss the potential of pharmacological chaperones (PCs) and enzymatic enhancers in the treatment of lysosomal and Parkinson disease. The review focuses on describing the biological activity of iminosugar-type compounds and non-carbohydrate activity enhancers. This research topic has been extensively investigated during the last decades but was poorly rewarded in terms of marketed drugs. The review is enjoyable to read and may be published in Pharmaceuticals after the following modifications/corrections.

We thank Reviewer 1 for having appreciated our review and having enjoyed its reading.

-A significant number of previous reviews focus on PC to treat lysosomal disorders, please specify the originality, different perspective provided by the present work.

We thank Reviewer 1 for having addressed this point. To our opinion, this review is particularly innovative since it is focused only on GCase and cover the most recent results, addressing also the potential use of the PCT for treating PD, which has not been extensively described yet. In addition, the second section dealing with the recent examples of allosteric chaperones is particularly innovative. To highlight this aspect, we added two sentences in the Introduction Section.

-An introductive figure presenting the maturation of glycolipids in the relevant cell compartments with the GCase activity (ER, Golgi, lysosomes) and the chaperoning activity of synthetic molecules may help the reader to better understand the general concept.

We thank Reviewer 1 for this suggestion: a representative cartoon, of the pharmacological chaperone activity was introduced and named Figure 1. The following figures were re-numbered accordingly.

-It seems that the in vitro enzymatic enhancement observed with iminosugar type PC is generally limited to 2-fold. In the review several examples of compounds giving this level of enhanced activity but at very different concentrations are provided. It is not clear for the reader at which concentration (micromolar, nanomolar?) a compound is considered as a good pharmacological chaperone in vitro. Please comment if possible.

This is a good point. However, this aspect strongly depends from the compound toxicity and from compound’s cell permeability, so it is difficult to make a general comment. In principle, a nanomolar concentration is better than a micromolar one, but if negligible toxicity is observed, also a compound which is active at a micromolar concentration might have a great potential in vivo. A sentence addressing this aspect has been added in the Conclusions.

-In the same vein, target selectivity is crucial when designing new iminosugars to limit detrimental side-effects. However, no indications of glycosidase selectivity is provided in the review.

This is also a good point. However, in our opinion, selectivity is important when proved on lysosomal enzymes (rather than commercial ones). We noticed that this was verified in most of the manuscripts cited in the Review. We therefore added the following sentence in the Conclusion section: “In most of the reports the selectivity towards GCase over other lysosomal glycosidases is also verified, which is important to avoid undesired side effects.”

-Structure-activity relationship is also poorly discussed. The GCase seems to accommodate a high diversity of iminosugars regardless of their cycle size and configuration of hydroxyl groups, making a rational design difficult to achieve. Presenting published structures of GCase-inhibitors, if any, may enrich the review.

We thank the Reviewer for addressing this issue. We decided not to include in this Review examples of potent GCase inhibitors that were not tested as PCs on mutated cells, since it happens that a too strong inhibition does not correspond to good chaperoning activity. This is clearly stated in the Introduction section and reiterated in the Conclusions section.

- The author presents the chaperoning activity of GCase with iminosugar-based compounds and an allosteric enhancement with non-carbohydrate derivatives. The situation may actually be more complex as these two types of carbohydrate mimics have been previously shown to be able to enhance the enzymatic activity of glycosidases (see:  Carbohydr.Res.2008, 343,2878–2886.) It may be of interest to more generally discuss the faculty of iminosugars to enhance the enzymatic activity in addition of their chaperoning effect.

We agree with the Reviewer that carbohydrate mimics can also act as enzyme enhancers towards commercial enzymes, as reported in Carbohydr.Res.2008, 343,2878–2886, or towards lysosomal enzymes, such as β-galactosidase with compound 78b of the Manuscript (Reference cited F. Stauffert et al. Org. Biomol. Chem. 2017,15, 3681-3705). To the best of our knowledge, however, there are no examples of iminosugars which act both as GCase enhancers (on wild-type enzyme) and as PC (on mutated cell lines). For this reason, we decided not to mention this aspect in this Review, which is focused on GCase.

-Reference 43 is actually not the first example of pharmacophores linked to iminosugars by CuAAc for GCase chaperoning, please see J. Org. Chem. 2011, 76, 19, 7757–7768.

We thank Reviewer 1 for citing this article, which previously employed the CuAAC to obtain a series of GCase PCs based on iminosugars bearing hydrophobic substituents. We therefore added this reference (numbered as Ref.43) in the text, before commenting the paper by Martínez-Bailén et al. (which is now Ref. 44). The following references were re-numbered accordingly.

-In the introduction to multivalent structures, rather than citing a mix of reviews and selected articles (ref 54-58), I would rather suggest to only cite all the relevant reviews published in the field (refs are missing).

Yes, the Reviewer is right. We separated Reviews from Research articles in the citations, and we added the missing references.

-Ref 59 “chemistry” should read “chemistry a European journal”

Thank you for noting this mistake. The reference has been corrected.

-Ref 62, is it really a significant multivalent effect in chaperoning evidenced in this study? If not the text should be pondered.

We apologize if the sentence was misleading, we intended that it was the first example of a multivalent PC for Fabry disease. However, we removed the world “unprecedented”. The new sentence is:

“More recently, some of us have described some multivalent derivatives with potential as pharmacological chaperones in fibroblasts of Fabry patients”

Line 511 ‘higher binding affinity for the catalytic pocket’ These compounds are suicidal inhibitors and it should be stated that the IC50 value observed is therefore not comparable with more conventional inhibitors.

The Reviewer is right, the sentence was misleading, we removed the whole sentence: The lowest IC50 values (1.24 nM and 1.94 nM, respectively) found for compounds 97 and 98 were in agreement with their stabilization effects. and higher binding affinity for the catalytic pocket”

Reviewer 2 Report

Martínez-Bailé M et al reviewed the pharmacological chaperone (PC) for glucocerebrosidase enzyme, a causative enzyme for Gaucher disease. These authors nicely introduced the historical background of PC development and its mode of action. Then, these authors described the structure-activity relationship of compounds tested for last 7 years. Finally, these authors discussed future perspective of the class of these compounds for Gaucher disease and other related sphingolipidoses of lysosomal storage disorders. In fact, this review also briefly covered an allosteric chaperone compounds which increase enzyme activity.

Major concerns:

1) These authors provide only an example of Gaucher mouse (Gba1D409/+) (Table 6). Please provide additional examples. If possible, please also show data of pharmacokinetics of the drugs to discuss further possibility of their clinical development.

Author Response

Martínez-Bailé M et al reviewed the pharmacological chaperone (PC) for glucocerebrosidase enzyme, a causative enzyme for Gaucher disease. These authors nicely introduced the historical background of PC development and its mode of action. Then, these authors described the structure-activity relationship of compounds tested for last 7 years. Finally, these authors discussed future perspective of the class of these compounds for Gaucher disease and other related sphingolipidoses of lysosomal storage disorders. In fact, this review also briefly covered an allosteric chaperone compounds which increase enzyme activity.

We thank the Reviewer 2 for the positive comment.

Major concerns:

  • These authors provide only an example of Gaucher mouse (Gba1D409/+) (Table 6). Please provide additional examples. If possible, please also show data of pharmacokinetics of the drugs to discuss further possibility of their clinical development.

Thank you for pointing this out.

Actually, the Gba1D409V/+ mouse model mentioned by Reviewer 2 is heterozygous for GD (healthy mouse carrier of GBA) and not homozygous for GD (a sick mouse). Sick mice models are more commonly employed for studying the mechanistic connection between GD and PD [Neurotoxicology 2009, 30, 1127-1132. Ann. Neurol. 2011, 69, 940-953. Proc. Natl. Acad. Sci. USA 2011, 108, 12101-12106.], rather than for investigating new drugs for PD, and this is not the focus of the review. To make it clearer, the caption of Table 6 was changed.

In addition, we provided another example of transgenic mice expressing the heterozygous L444P mutation in the murine GBA gene.

As required by the Reviewer, we also included in the text the pharmacokinetic properties for compound S-181 (110). Regarding IFG (1) and ABX (99), a sentence explaining that the measurement of efficacy in the mice brain evidences their ability to cross the blood-brain barrier (BBB) was added, together with a final comment.

We removed a row in Table 6 dealing with studies of ABX (99) on wild-type primates, because it is not a disease model.
